# InsActor: Instruction-driven Physics-based Characters

**Jiawei Ren**[*1]    **Mingyuan Zhang**[*1]    **Cunjun Yu**[*2]    **Xiao Ma**[3]    **Liang Pan**[1]    **Ziwei Liu**[1]

[1] S-Lab, Nanyang Technological University
[2] National University of Singapore
[3] Dyson Robot Learning Lab

## Abstract

Generating animation of physics-based characters with intuitive control has long been a desirable task with numerous applications. However, generating physically simulated animations that reflect high-level human instructions remains a difficult problem due to the complexity of physical environments and the richness of human language. In this paper, we present **InsActor**, a principled generative framework that leverages recent advancements in diffusion-based human motion models to produce instruction-driven animations of physics-based characters. Our framework empowers InsActor to capture complex relationships between high-level human instructions and character motions by employing diffusion policies for flexibly conditioned motion planning. To overcome invalid states and infeasible state transitions in planned motions, InsActor discovers low-level skills and maps plans to latent skill sequences in a compact latent space. Extensive experiments demonstrate that InsActor achieves state-of-the-art results on various tasks, including instruction-driven motion generation and instruction-driven waypoint heading. Notably, the ability of InsActor to generate physically simulated animations using high-level human instructions makes it a valuable tool, particularly in executing long-horizon tasks with a rich set of instructions. Our project page is available at jiawei-ren.github.io/projects/insactor/index.html

## 1   Introduction

Generating life-like natural motions in a simulated environment has been the focus of physics-based character animation [24, 16]. To enable user interaction with the generated motion, various conditions such as waypoints have been introduced to control the generation process [40, 15]. In particular, human instructions, which have been widely adopted in text generation and image generation, have recently drawn attention in physics-simulated character animation [15]. The accessibility and versatility of human instructions open up new possibilities for downstream physics-based character applications.

Therefore, we investigate a novel task in this work: generating physically-simulated character animation from human instruction. The task is challenging for existing approaches. While motion tracking [4] is a common approach for character animation, it presents challenges when tracking novel motions generated from free-form human language. Recent advancements in language-conditioned controllers [15] have demonstrated the feasibility of managing characters using instructions, but they struggle with complex human commands. On the other hand, approaches utilizing conditional generative models to directly generate character actions [14] fall short of ensuring the accuracy necessary for continuous control.

To tackle this challenging task, we present **InsActor**, a framework that employs a hierarchical design for creating instruction-driven, physics-based characters. At the high level, InsActor generates motion

---

[*]Equal contribution

37th Conference on Neural Information Processing Systems (NeurIPS 2023).

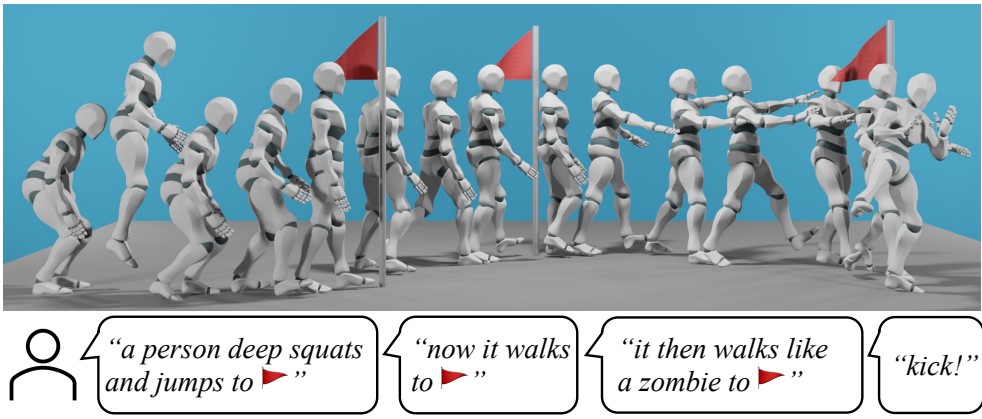

Figure 1: **InsActor** enables controlling physics-based characters with human instructions and intended target position. The figure illustrates this by depicting several "flags" on a 2D plane, each representing a relative target position such as (0,2) starting from the origin.

plans conditioned on human instructions. This approach enables the seamless integration of human commands, resulting in more coherent and intuitive animations. To accomplish this, InsActor utilizes a diffusion policy [23, 5] to generate actions in the joint space conditioned on human inputs. It allows flexible test-time conditioning, which can be leveraged to complete novel tasks like waypoint heading without task-specific training. However, the high-level diffusion policy alone does not guarantee valid states or feasible state transitions, making it insufficient for direct execution of the plans using inverse dynamics [1]. Therefore, at the low level, InsActor incorporates unsupervised skill discovery to handle state transitions between pairs of states, employing an encoder-decoder architecture. Given the state sequence in joint space from the high-level diffusion policy, the low-level policy first encodes it into a compact latent space to address any infeasible joint actions from the high-level diffusion policy. Each state transition pair is mapped to a skill embedding within this latent space. Subsequently, the decoder translates the embedding into the corresponding action. This hierarchical architecture effectively breaks down the complex task into two manageable tasks at different levels, offering enhanced flexibility, scalability, and adaptability compared to existing solutions.

Given that InsActor generates animations that inherently ensure physical plausibility, the primary evaluation criteria focus on two aspects: fidelity to human instructions and visual plausibility of the animations. Through comprehensive experiments assessing the quality of the generated animations, InsActor demonstrates its ability to produce visually captivating animations that faithfully adhere to instructions, while maintaining physical plausibility. Furthermore, thanks to the flexibility of the diffusion model, animations can be further customized by incorporating additional conditions, such as waypoints, as illustrated in Figure 1, showcasing the broad applicability of InsActor. In addition, InsActor also serves as an important baseline for language conditioned physics-based animation generation.

## 2 Related Works

### 2.1 Human Motion Generation

Human motion generation aims to produce versatile and realistic human movements [12, 21, 22, 39]. Recent advancements have enabled more flexible control over motion generation [2, 10, 38, 9]. Among various methods, the diffusion model has emerged as a highly effective approach for generating language-conditioned human motion [36, 44]. However, ensuring physical plausibility, such as avoiding foot sliding, remains challenging due to the absence of physical priors and interaction with the environment [42, 29, 32]. Some recent efforts have attempted to address this issue by incorporating physical priors into the generation models [41], such as foot contact loss [36]. Despite this progress, these approaches still struggle to adapt to environmental changes and enable interaction with the environment. To tackle these limitations, we propose a general framework for generating long-horizon human animations that allow characters to interact with their environment and remain

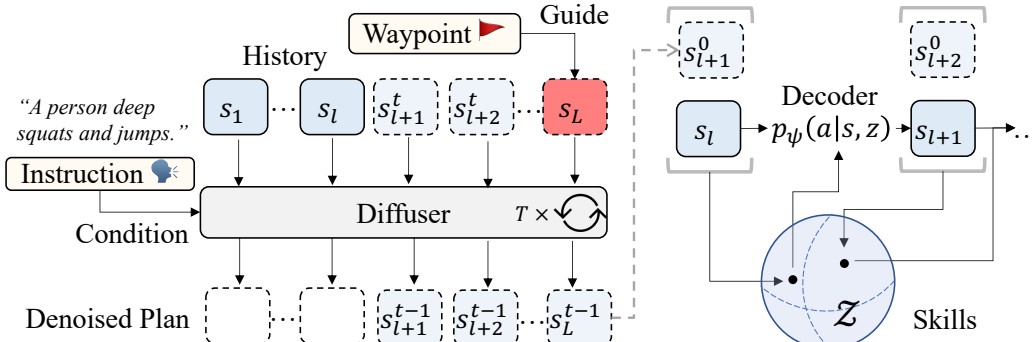

Figure 2: The overall framework of InsActor. At the high level, the diffusion model generates state sequences from human instructions and waypoint conditions. At the low level, each state transition is encoded into a skill embedding in the latent space and decoded to an action.

robust to environmental changes. Our approach strives to bridge the gap between understanding complex high-level human instructions and generating physically-simulated character motions.

## 2.2 Language-Conditioned Control

Language-Conditioned Control aims to guide an agent's behavior using natural language, which has been extensively applied in physics-based animation and robot manipulation [35, 20, 19] to ensure compliance with physical constraints. However, traditional approaches often necessitate dedicated language modules to extract structured expressions from free-form human languages or rely on handcrafted rules for control [33, 43, 34]. Although recent attempts have trained data-driven controllers to generate actions directly from human instructions [18, 15], executing long-horizon tasks remains challenging due to the need to simultaneously understand environmental dynamics, comprehend high-level instructions, and generate highly accurate control. The diffusion model, considered one of the most expressive models, has been introduced to generate agent actions [14].

Nonetheless, current methods are unable to accurately control a humanoid character, as evidenced by our experiments. Recent work utilizing diffusion model to generate high-level pedestrian trajectories and ground the trajectories with a low level controller [30]. Compared with existing works, InsActor employs conditional motion generation to capture intricate relationships between high-level human instructions and character motions beyond pedestrian trajectories, and subsequently deploys a low-level skill discovery incorporating physical priors. This approach results in animations that are both visually striking and physically realistic.

## 3 The Task of Instruction-driven Physics-based Character Animation

We formulate the task as conditional imitation learning, in which we learn a goal-conditioned policy that outputs an action $\mathbf{a} \in A$ based on the current state $\mathbf{s} \in S$ and an additional condition $\mathbf{c} \in C$ describing the desired character behavior. The environment dynamics are represented by the function $\mathcal{T} : S \times A \to S$.

The task state comprises the character's pose and velocity, including position $\mathbf{p}$, rotation $\mathbf{q}$, linear velocity $\dot{\mathbf{p}}$, and angular velocity $\dot{\mathbf{q}}$ for all links in local coordinates. Consequently, the task state, $\mathbf{s}$, contains this information: $\mathbf{s} := \{\mathbf{p}, \mathbf{q}, \dot{\mathbf{p}}, \dot{\mathbf{q}}\}$. Following common practice, we employ PD controllers to drive the character. Given the current joint angle $\mathbf{q}$, angular velocity $\dot{\mathbf{q}}$, and target angle $\tilde{\mathbf{q}}$, the torque on the joint actuator is computed as $k_p(\tilde{\mathbf{q}} - \mathbf{q}) + k_d(\dot{\tilde{\mathbf{q}}} - \dot{\mathbf{q}})$, where $\dot{\tilde{\mathbf{q}}} = 0$, $k_p$ and $k_d$ are manually specified PD controller gains. We maintain $k_p$ and $k_d$ identical to the PD controller used in DeepMimic [25]. The action involves generating target angles for all joints to control the character.

Our work addresses a realistic setting in which demonstration data consists solely of states without actions, a common scenario in motion datasets collected from real humans where obtaining actions is challenging [26, 3, 13]. To train our model, we use a dataset of trajectory-condition pairs $\mathcal{D} = \{(\boldsymbol{\tau}^i, \boldsymbol{c}^i)\}_{i=1}^N$, where $\boldsymbol{\tau} = \{\mathbf{s}_1^*, ..., \mathbf{s}_L^*\}$ denotes a state-only demonstration generated by the expert of

length $L$. We use $\mathbf{s}^*$ to denote states from the expert demonstration. For example, a human instruction can be "walk like a zombie" and the trajectory would be a state sequence describing the character's motion. Our objective is to learn a policy in conjunction with environment dynamics $\mathcal{T}$, which can replicate the expert's trajectory for given instruction $\mathbf{c} \in C$.

## 4 The Framework of InsActor

The proposed method, InsActor, employs a unified hierarchical approach for policy learning, as depicted in Figure 2. Initially, a diffusion policy interprets high-level human instructions, generating a sequence of actions in the joint space. In our particular case, the action in the joint space can be regarded as the state of the animated character. Subsequently, each pair of actions in the joint space are mapped into the corresponding skill embedding in the latent space, ensuring their plausibility while producing desired actions for character control in accordance with motion priors. Consequently, InsActor effectively learns intricate policies for animation generation that satisfy user specifications in a physically-simulated environment. The inference process of InsActor is detailed in Algorithm 1.

### 4.1 High-Level Diffusion Policy

For the high-level state diffusion policy, we treat the joint state of the character as its action. We follow the state-of-the-art approach of utilizing diffusion models to carry out conditional motion generation [44, 36]. We denote the human instruction as $c$ and the state-only trajectory as $\tau$.

**Trajectory Curation.** In order to use large-scale datasets for motion generation in a physical simulator, it is necessary to retarget the motion database to a simulated character to obtain a collection of reference trajectories. Large-scale text-motion databases HumanML3D [8] and KIT-ML [26] use SMPL [17] sequences to represent motions. SMPL describes both the body shape and the body poses, where the body poses include pelvis location and rotation, the relative joint rotation of the 21 body joints. We build a simulated character to have the same skeleton as SMPL. We scale the simulated character to have a similar body size to a mean SMPL neutral shape. For retargeting, we directly copy the joint rotation angle, pelvis rotation, and translation to the simulated character. A vertical offset is applied to compensate for different floor heights.

**Diffusion Models.** Diffusion models [11] are probabilistic techniques used to remove Gaussian noise from data and generate a clean output. These models consist of two processes: the diffusion process and the reverse process. The diffusion process gradually adds Gaussian noise to the original data for a specified number of steps, denoted by $T$, until the distribution of the noise closely approximates a standard Gaussian distribution, denoted by $\mathcal{N}(\mathbf{0}, \mathbf{I})$. This generates a sequence of noisy trajectories denoted by $\tau_{1:T} = \{\tau_1, ..., \tau_T\}$. The original data is sampled from a conditional distribution, $\tau_0 \sim p(\tau_0 \mid c)$, where $c$ is the instruction. Assuming that the variance schedules are determined by $\beta_t$, the diffusion process is defined as:

$$q(\tau_{1:T}|\tau_0) := \prod_{t=1}^{T} q(\tau_t|\tau_{t-1}), \quad q(\tau_t|\tau_{t-1}) := \mathcal{N}(\tau_t; \sqrt{1 - \beta_t}\tau_{t-1}, \beta_t \mathbf{I}), \quad (1)$$

where $q(\tau_t|\tau_{t-1})$ is the conditional distribution of each step in the Markov chain. The parameter $\beta_t$ controls the amount of noise added at each step $t$, with larger values resulting in more noise added. The reverse process in diffusion models is another Markov chain that predicts and removes the added noise using a learned denoising function. In particular, we encode language $c$ into an encoded latent vector, $\hat{c}$, using the classical transformer [37] as the language encoder, $\hat{c} = \mathcal{E}(c)$. Thus, the reverse process starts with a distribution $p(\tau_T) := \mathcal{N}(\tau_T; \mathbf{0}, \mathbf{I})$ and is defined as:

$$p(\tau_{0:T}|\hat{c}) := p(\tau_T)\prod_{t=1}^{T} p(\tau_{t-1}|\tau_t, \hat{c}), \quad p(\tau_{t-1}|\tau_t, \hat{c}) := \mathcal{N}(\tau_{t-1}; \mu(\tau_t, t, \hat{c}), \Sigma(\tau_t, t, \hat{c})).$$
$$(2)$$

Here, $p(\tau_{t-1}|\tau_t, \hat{c})$ is the conditional distribution at each step in the reverse process. The mean and covariance of the Gaussian are represented by $\mu$ and $\Sigma$, respectively. During training, steps $t$ are uniformly sampled for each ground truth motion $\tau_0$, and a sample is generated from $q(\tau_t|\tau_0)$. Instead of predicting the noise term $\epsilon$ [11], the model predicts the original data $\tau_0$ directly which has

---

**Algorithm 1** Inference of InsActor

---

1: **Input:** A instruction $\boldsymbol{c}$, the diffusion model $f_\theta$, the skill encoder, $q_\phi$ and decoder $p_\psi$. Diffusion steps $T$. A language encoder $\mathcal{E}$. A history $o = \{\hat{\mathbf{s}}_1, ..., \hat{\mathbf{s}}_l\}$. A waypoint $h$. Animation length $L$.
2: **Output:** A physically simulated trajectory, $\hat{\boldsymbol{\tau}}$.
   ▷ Generate state sequence.
3: $w \leftarrow$ Initialize a plan from a Gaussian noise, $w \sim \mathcal{N}(\mathbf{0}, \mathbf{I})$
4: $w \leftarrow$ Apply the inpainting strategy in [14] for history $o$ and waypoint $h$ to guide diffusion.
5: $\hat{\boldsymbol{c}} \leftarrow$ Encode the instruction $\mathcal{E}(\boldsymbol{c})$
6: $\boldsymbol{\tau} = \{\mathbf{s}_1, ....\mathbf{s}_L\} \leftarrow$ Generate trajectory with the diffusion model $f_\theta(w, T, \hat{\boldsymbol{c}})$
   ▷ Generate action sequence in a closed-loop manner.
7: **for** $i = l, l+1, ..., (L-1)$ **do**
8: $\quad \boldsymbol{z}_i \leftarrow$ Sample $\boldsymbol{z}$ from $q_\phi(\boldsymbol{z}_i | \hat{\mathbf{s}}_i, \mathbf{s}_{i+1})$
9: $\quad \boldsymbol{a}_i \leftarrow$ Sample action form $p_\psi(\boldsymbol{a}_i | \hat{\mathbf{s}}_i, \boldsymbol{z}_i)$
10: $\quad \hat{\mathbf{s}}_{i+1} \leftarrow$ Get next state $\mathcal{T}(\hat{\mathbf{s}}_{i+1} | \hat{\mathbf{s}}_i, \boldsymbol{a}_i)$
11: **end for**
12: Output $\hat{\boldsymbol{\tau}} = \{\hat{\mathbf{s}}_{l+1}, ....\hat{\mathbf{s}}_L\}$

---

the equivalent formulation [28, 36]. This is done by using a neural network $f_\theta$ parameterized by $\theta$ to predict $\boldsymbol{\tau}_0$ from the noisy trajectory $\boldsymbol{\tau}_t$ and the condition $\hat{c}$ at each step $t$ in the denoise process. The model parameters are optimized by minimizing the mean squared error between the predicted and ground truth data using the loss function:

$$\mathcal{L}_{\text{Plan}} = \mathrm{E}_{t \in [1,T], \boldsymbol{\tau}_0 \sim p(\boldsymbol{\tau}_0 | \boldsymbol{c})}[\| \boldsymbol{\tau}_0 - f_\theta(\boldsymbol{\tau}_t, t, \hat{\boldsymbol{c}}) \|], \tag{3}$$

where $p(\boldsymbol{\tau}_0 | \boldsymbol{c})$ is the conditional distribution of the ground truth data, and $\| \cdot \|$ denotes the mean squared error. By directly predicting $\boldsymbol{\tau}_0$, this formulation avoids repeatedly adding noise to $\boldsymbol{\tau}_0$ and is more computationally efficient.

**Guided Diffusion.** Diffusion models allow flexible test-time conditioning through guided sampling, for example, classifier-guided sampling [6]. Given an objective function as a condition, gradients can be computed to optimize the objective function and perturb the diffusion process. In particular, a simple yet effective *inpainting* strategy can be applied to introduce state conditions [14], which is useful to generate a plan that adheres to past histories or future goals. Concretely, the inpainting strategy formulates the state conditioning as a Dirac delta objective function. Optimizing the objective function is equivalent to directly presetting noisy conditioning states and inpainting the rest. We leverage the inpainting strategy to achieve waypoint heading and autoregressive generation.

**Limitation.** Despite the ability to model complex language-to-motion relations, motion diffusion models can generate inaccurate low-level details, which lead to physically implausible motions and artifacts like foot floating and foot penetration [36]. In the context of state diffusion, the diffuser-generated states can be invalid and the state transitions can be infeasible. Thus, direct tracking of the diffusion plan can be challenging.

## 4.2 Low-Level Skill Discovery

To tackle the aforementioned challenge, we employ low-level skill discovery to safeguard against unexpected states in poorly planned trajectories. Specifically, we train a Conditional Variational Autoencoder to map state transitions to a compact latent space in an unsupervised manner [40]. This approach benefits from a repertoire of learned skill embedding within a compact latent space, enabling superior interpolation and extrapolation. Consequently, the motions derived from the diffusion model can be executed by natural motion primitives.

**Skill Discovery.** Assuming the current state of the character is $\hat{\mathbf{s}}_l$, the first step in constructing a compact latent space for skill discovery is encoding the state transition in a given reference motion sequence, $\boldsymbol{\tau}_0 = \{\mathbf{s}_1^*, ..., \mathbf{s}_L^*\}$, into a latent variable, $\boldsymbol{z}$, and we call this skill embedding. This variable represents a unique skill required to transition from $\hat{\mathbf{s}}_l$ to $\mathbf{s}_{l+1}^*$. The neural network used to encode the skill embedding is referred to as the encoder, $q_\phi$, parameterized by $\phi$, which produces a Gaussian distribution:

$$q_\phi(\boldsymbol{z}_l | \hat{\mathbf{s}}_l, \mathbf{s}_{l+1}^*) := \mathcal{N}(\boldsymbol{z}_l; \mu_\phi(\hat{\mathbf{s}}_l, \mathbf{s}_{l+1}^*), \Sigma_\phi(\hat{\mathbf{s}}_l, \mathbf{s}_{l+1}^*)), \tag{4}$$

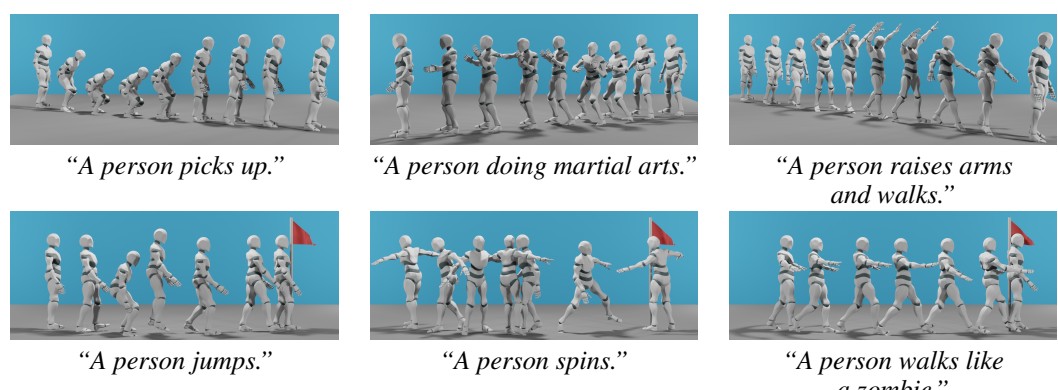

*"A person picks up."*     *"A person doing martial arts."*     *"A person raises arms and walks."*

*"A person jumps."*     *"A person spins."*     *"A person walks like a zombie."*

Figure 3: **Qualitative results of InsActor with corresponding instructions. Top:** only human instruction. **Bottom:** human instruction and waypoint target.

where $\mu_\phi$ is the mean and $\Sigma_\phi$ is the isotropic covariance matrix. Once we obtain the latent variable, a decoder, $p_\psi(\boldsymbol{a}_l|\hat{\mathbf{s}}_l, \boldsymbol{z}_l)$, parameterized by $\psi$, generates the corresponding actions, $\boldsymbol{a}$, by conditioning on the latent variable $\boldsymbol{z}_l$ and the current state $\hat{\mathbf{s}}_l$:

$$\boldsymbol{a}_l \sim p_\psi(\boldsymbol{a}_l|\hat{\mathbf{s}}_l, \boldsymbol{z}_l) \tag{5}$$

Subsequently, using the generated action, the character transitions into the new state, $\hat{\mathbf{s}}_{l+1}$, via the transition function $\mathcal{T}(\hat{\mathbf{s}}_{l+1}|\hat{\mathbf{s}}_l, \boldsymbol{a}_l)$. By repeating this process, we can gather a generated trajectory, denoted as $\hat{\boldsymbol{\tau}} = \{\hat{\mathbf{s}}_1, ..., \hat{\mathbf{s}}_L\}$. The goal is to mimic the given trajectory $\boldsymbol{\tau}_0$ by performing the actions. Thus, to train the encoder and decoder, the main supervision signal is derived from the difference between the resulting trajectory $\hat{\boldsymbol{\tau}}$ and the reference motion, $\boldsymbol{\tau}_0$.

**Training.** Our approach leverages differentiable physics to train the neural network end-to-end without the need for a separate world model [31]. This is achieved by implementing the physical laws of motion as differentiable functions, allowing the gradient to flow through them during backpropagation. Concretely, by executing action $\boldsymbol{a}_l \sim p_\psi(\boldsymbol{a}_l|\hat{\mathbf{s}}_l, \boldsymbol{z}_l)$ at state $\hat{\mathbf{s}}_l$, the induced state $\hat{\mathbf{s}}_{l+1}$ is differentiable with respect to the policy parameter $\psi$ and $\phi$. Thus, directly minimizing the difference between the predicted state $\hat{\mathbf{s}}$ and the $\mathbf{s}^*$ gives an efficient and effective way of training an imitation learning policy [31]. The Brax [7] simulator is used due to its efficiency and easy parallelization, allowing for efficient skill discovery. It also ensures that the learned skill is trained on the actual physical environment, rather than a simplified model of it, leading to a more accurate and robust representation.

Thus, the encoder-decoder is trained with an objective that minimizes the discrepancy between resulting trajectories and Kullback–Leibler divergence between the encoded latent variable and the prior distribution, which is a standard Gaussian,

$$\mathcal{L}_{\text{Skill}} = \| \boldsymbol{\tau}_0 - \hat{\boldsymbol{\tau}} \| + \lambda D_{\text{KL}}(q_\phi(\boldsymbol{z}|\mathbf{s}, \mathbf{s}') \| \mathcal{N}(\mathbf{0}, \mathbf{I})), \tag{6}$$

where $\| \cdot \|$ denotes the mean squared error and $(\mathbf{s}, \mathbf{s}')$ is a pair of states before and after transition. The latter term encourages the latent variables to be similar to the prior distribution, ensuring the compactness of the latent space. $\lambda$ is the weight factor that controls the compactness. During inference, we map the generated state sequence from the diffusion model to the skill space to control the character.

## 5 Experiments

The goal of our experiment is to evaluate the effectiveness and robustness of InsActor in generating physically-simulated and visually-natural character animations based on high-level human instructions. Specifically, we aim to investigate *1)* whether InsActor can generate animations that adhere to human instructions while being robust to physical perturbations, *2)* whether InsActor can accomplish waypoint heading while being faithful to the language descriptions, and *3)* the impact of several design choices, including the hierarchical design and the weight factor for skill space compactness.

Table 1: **Quantitative results on the KIT-ML test set.** †: with perturbation.

| Methods | R Precision↑ | | | Multimodal Dist↓ | FID↓ | Diversity↑ |
|---|---|---|---|---|---|---|
| | Top 1 | Top 2 | Top 3 | | | |
| DReCon [4] | $0.243\pm.000$ | $0.420\pm.021$ | $0.522\pm.039$ | $2.310\pm.097$ | $1.055\pm.162$ | $4.259\pm.014$ |
| PADL [15] | $0.091\pm.003$ | $0.172\pm.008$ | $0.242\pm.015$ | $3.482\pm.038$ | $3.889\pm.104$ | $2.940\pm.031$ |
| InsActor (Ours) | $\mathbf{0.352}\pm.013$ | $\mathbf{0.550}\pm.010$ | $\mathbf{0.648}\pm.015$ | $\mathbf{1.808}\pm.027$ | $\mathbf{0.786}\pm.055$ | $\mathbf{4.392}\pm.071$ |
| DReCon† [4] | $0.253\pm.013$ | $0.384\pm.006$ | $0.447\pm.006$ | $2.764\pm.003$ | $1.973\pm.100$ | $4.252\pm.040$ |
| PADL† [15] | $0.100\pm.012$ | $0.158\pm.011$ | $0.217\pm.015$ | $3.783\pm.069$ | $4.706\pm.298$ | $3.168\pm.065$ |
| InsActor (Ours)† | $\mathbf{0.323}\pm.013$ | $\mathbf{0.496}\pm.017$ | $\mathbf{0.599}\pm.008$ | $\mathbf{2.147}\pm.061$ | $\mathbf{1.043}\pm.091$ | $\mathbf{4.359}\pm.073$ |

Table 2: **Quantitative results on the HumanML3D test set.** †: with perturbation.

| Methods | R Precision↑ | | | Multimodal Dist↓ | FID↓ | Diversity↑ |
|---|---|---|---|---|---|---|
| | Top 1 | Top 2 | Top 3 | | | |
| DReCon [4] | $0.265\pm.007$ | $0.391\pm.004$ | $0.470\pm.001$ | $2.570\pm.002$ | $1.244\pm.040$ | $4.070\pm.062$ |
| PADL [15] | $0.144\pm.003$ | $0.227\pm.012$ | $0.297\pm.018$ | $3.349\pm.030$ | $2.162\pm.022$ | $3.736\pm.091$ |
| InsActor (Ours) | $\mathbf{0.331}\pm.000$ | $\mathbf{0.497}\pm.015$ | $\mathbf{0.598}\pm.001$ | $\mathbf{1.971}\pm.004$ | $\mathbf{0.566}\pm.023$ | $\mathbf{4.165}\pm.076$ |
| DReCon† [4] | $0.233\pm.000$ | $0.352\pm.001$ | $0.424\pm.004$ | $2.850\pm.002$ | $1.829\pm.002$ | $4.008\pm.147$ |
| PADL† [15] | $0.117\pm.005$ | $0.192\pm.003$ | $0.254\pm.000$ | $3.660\pm.040$ | $2.964\pm.115$ | $3.849\pm.159$ |
| InsActor (Ours)† | $\mathbf{0.312}\pm.001$ | $\mathbf{0.455}\pm.006$ | $\mathbf{0.546}\pm.003$ | $\mathbf{2.203}\pm.006$ | $\mathbf{0.694}\pm.005$ | $\mathbf{4.212}\pm.154$ |

## 5.1 Implementation Details

To implement the experiment, we use Brax [7] to build the environment and design a simulated character based on DeepMimic [25]. The character has 13 links and 34 degrees of freedom, weighs 45kg, and is 1.62m tall. Contact is applied to all links with the floor. For details of neural network architecture and training, we refer readers to the supplementary materials.

## 5.2 Evaluation Protocols

**Datasets.** We use two large scale text-motion datasets, KIT-ML [26] and HumanML3D [8], for training and evaluation. KIT-ML has 3,911 motion sequences and 6,353 sequence-level language descriptions, HumanML3D provides 44,970 annotations on 14,616 motion sequences. We adopt the original train/test splits in the two datasets.

**Metrics.** We employ the following evaluation metrics:

1. *R Precision*: For every pair of generated sequence and instruction, we randomly pick 31 additional instructions from the test set. Using a trained contrastive model, we then compute the average top-k accuracy.

2. *Frechet Inception Distance (FID):* We use a pre-trained motion encoder to extract features from both the generated animations and ground truth motion sequences. The FID is then calculated between these two distributions to assess their similarity.

3. *Multimodal Distance:* With the help of a pre-trained contrastive model, we compute the disparity between the text feature derived from the given instruction and the motion feature from the produced animation. We refer to this as the multimodal distance.

4. *Diversity:* To gauge diversity, we randomly divide the generated animations for all test texts into pairs. The average joint differences within each pair are then computed as the metric for diversity.

5. *Success Rate:* For waypoint heading tasks, we compute the Euclidean distance between the final horizontal position of the character pelvis and the target horizontal position. If the distance is less than 0.5 m, we deem it a success. We perform each evaluation three times and report the statistical interval with 95% confidence.

Table 3: **Quantitative results for the waypoint heading task.** Evaluated on HumanML3D. We set the start point at (0,0) and the waypoint uniformly sampled from a 6x6 square centered at (0,0). It is considered a successful waypoint heading if the final position is less than 0.5m away from the waypoint. **L**: Langauge. **W**: Waypoint.

| Method | L | W | R Precision↑ Top 3 | Multimodal Dist↓ | FID↓ | Diversity↑ | Success Rate↑ |
|---|---|---|---|---|---|---|---|
| DReCon [4] | ✓ | ✓ | 0.178±.000 | 4.192±.019 | 8.607±.114 | 2.583±.157 | 0.380±.002 |
| InsActor (Ours) | ✗ | ✓ | 0.089±.001 | 4.106±.001 | 3.041±.101 | 3.137±.029 | **0.935**±.002 |
| InsActor (Ours) | ✓ | ✗ | **0.598**±.001 | **1.971**±.004 | **0.566**±.023 | **4.165**±.076 | 0.081±.004 |
| InsActor (Ours) | ✓ | ✓ | 0.388±.003 | 2.753±.009 | 2.527±.015 | 3.285±.034 | 0.907±.002 |

Table 4: **Ablation on hierarchical design.** Evaluated on the KIT-ML test set.

| High | Low | R Precision↑ | | | MultiModal Dist↓ | FID↓ | Diversity↑ |
|---|---|---|---|---|---|---|---|
| | | Top 1 | Top 2 | Top 3 | | | |
| ✗ | ✓ | 0.264±.011 | 0.398±.016 | 0.460±.018 | 2.692±.034 | 1.501±.095 | 4.370±.066 |
| ✓ | ✗ | 0.068±.011 | 0.145±.030 | 0.188±.024 | 3.707±.096 | 1.106±.093 | 4.148±.098 |
| ✓ | ✓ | **0.352**±.013 | **0.550**±.010 | **0.648**±.015 | **1.808**±.027 | **0.786**±.055 | **4.392**±.071 |

## 5.3 Comparative Studies for Instruction-driven Character Animation

**Comparison Methods.** We compare InsActor with two baseline approaches: *1)* DReCon [4]. We adapted the responsive controller framework from DReCon [4]. We use the diffuser as a kinematic controller and train a target-state tracking policy. The baseline can also be viewed as a Decision Diffuser [1] with a long planning horizon, where a diffuser plans the future states and a tracking policy solves the inverse dynamics. *2)* PADL [15]: We adapt the language-conditioned control policy in PADL [15], where language instructions are encoded by a pretrained cross-modal text encoder [27] and input to a control policy that directly predict actions. It is also a commonly used learning paradigm in conditional imitation learning [34]. Since the two baselines have no publicly available implementations, we reproduce them and train the policies with DiffMimic [31].

**Settings.** We utilize two different settings to assess InsActor's robustness. In the first setting, we evaluate the models in a clean, structured environment devoid of any perturbation. In the second setting, we introduce perturbations by spawning a 2kg box to hit the character every 1 second, thereby evaluating whether the humanoid character can still adhere to human instructions even when the environment changes.

**Results.** We present the results in Table 1 and Table 2 and qualitative results in Figure 3. Compared to the dataset used in PADL that consists of 131 motion sequences and 256 language captions, our benchmark dataset is two orders larger, *where the language-conditioned single-step policy used in PADL has difficulty to scaling up*. In particular, the inferior performance in language-motion matching metrics suggests that a single-step policy fails to understand unseen instructions and model the many-to-many instruction-motion relation. Compared to PADL, DReCon shows a better result in language-motion matching thanks to the high-level motion planning. However, unlike Motion Matching used in DReCon that produces high-quality kinematic motions, *the diffuser generates invalid states and infeasible state transitions, which fails DReCon's tracking policy and results in a low FID*. In comparison, InsActor significantly outperforms the two baselines on all metrics. Moreover, the experiment reveals that environmental perturbations do not significantly impair the performance of InsActor, showcasing InsActor's robustness.

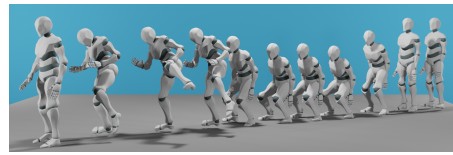 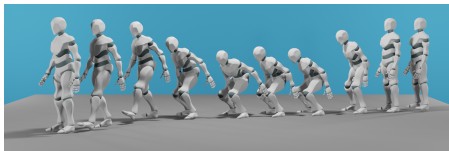

*"A person crouches."* +
*"A person kicks."*

*"A person crouches."* +
*"A person runs."*

Figure 4: **Qualitative results of InsActor with history conditioning.** Generation is conditioned on the second human instruction and history motion.

## 5.4 Instruction-driven Waypoint Heading

**Waypoint Heading.** Thanks to the flexibility of the diffusion model, InsActor can readily accomplish the waypoint heading task, a common task in physics-based character animation [40, 15]. This task necessitates the simulated character to move toward a target location while complying with human instructions. For instance, a human instruction might be, "walk like a zombie." In this case, the character should navigate toward the target position while mimicking the movements of a zombie.

**Guided Diffusion.** We accomplish this using guided diffusion. Concretely, we adopt the inpainting strategy in Diffuser [14]. Prior to denoising, we replace the Gaussian noise in the first and last 25% frames with the noisy states of the character standing at the starting position and target position respectively.

**Results.** We conduct this experiment with the model trained on HumanML3D. We contrast InsActor with DReCon and two InsActor variants: *1)* InsActor without language condition, and *2)* InsActor without targeting. Our experimental results demonstrate that InsActor can effectively accomplish the waypoint heading task by leveraging guided diffusion. The model trained on HumanML3D is capable of moving toward the target position while following the given human instructions, as evidenced by a low FID score and high precision. Although adding the targeting position condition to the diffusion process slightly compromises the quality of the generated animation, the outcome is still satisfactory. Moreover, the success rate of reaching the target position is high, underscoring the effectiveness of guided diffusion. Comparing InsActor with its two variants highlights the importance of both the language condition and the targeting in accomplishing the task. Comparing InsActor with DReCon shows the importance of skill mapping, particularly when more infeasible state transitions are introduced by the waypoint guiding. Without skill mapping, DReCon only has a 38.0% success rate, which drops drastically from the 90.7% success rate of InsActor.

**Multiple Waypoints.** Multiple waypoints allow users to interactively instruct the character. We achieve this by autoregressively conditioning the diffusion process to the history motion, where a qualitative result is shown in Figure 4. Concretely, we inpaint the first 25% with the latest history state sequences. We show qualitative results for multiple-waypoint following in Figure 1 and more in the supplementary materials.

## 5.5 Ablation Studies

**Hierarchical Design.** To understand the importance of the hierarchical design in this task, we performed an ablation study on its structure. We compared our approach to two baselines: *1)* A policy with only a high-level policy, wherein the diffusion model directly outputs the skills, analogous to the *Diffuser* approach [14]; *2)* A low-level policy that directly predicts single-step skills. We show the results in Table 4. By leveraging skills, the low-level policy improves from PADL but still grapples with comprehending the instructions due to the absence of language understanding. Conversely, without the low-level policy, the skills generated directly by the diffusion model are of poor precision. Although the use of skills safeguards the motions to be natural and score high in FID, the error accumulation deviates the plan from the language description and results in a low R-precision. The experimental results underscore the efficacy of the hierarchical design of InsActor.

**Weight Factor.** InsActor learns a compact latent space for skill discovery to overcome infeasible plans generated by the diffusion model. We conduct an ablation study on the weight factor, $\lambda$, which controls the compactness of the skill space. Our findings suggest that a higher weight factor results in a more compact latent space, however, it also curtails the instruction-motion alignment. Conversely, a lower weight factor permits a greater diversity in motion generation, but it might also lead to less plausible and inconsistent motions. Hence, it is vital to find a balance between these two factors to optimize performance for the specific task at hand.

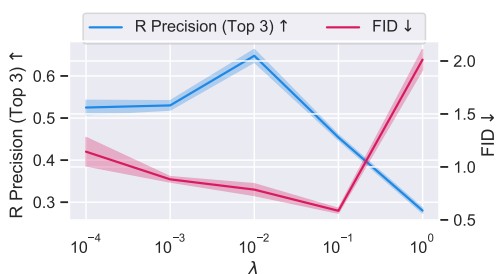

Figure 5: **Ablation study on the weight factor $\lambda$.** Evaluated on the KIT-ML dataset.

## 6 Conclusion

In conclusion, we have introduced InsActor, a principled framework for physics-based character animation generation from human instructions. By utilizing a diffusion model to interpret language instructions into motion plans and mapping them to latent skill vectors, InsActor can generate flexible physics-based animations with various and mixed conditions including waypoints. We hope InsActor would serve as an important baseline for future development of instruction-driven physics-based animation. While InsActor is capable of generating such animations, there are crucial but exciting challenges ahead. One limitation is the computational complexity of the diffusion model, which may pose challenges for scaling up the approach to more complex environments and larger datasets. Additionally, the current version of InsActor assumes access to expert demonstrations for training, which may limit its applicability in real-world scenarios where such data may not be readily available. Furthermore, while InsActor is capable of generating physically-reliable and visually-plausible animations, there is still room for improvement in terms of the quality and diversity of generated animations. There are mainly two aspects for future development, improving the quality of the differentiable physics for more realistic simulations and enhancing the expressiveness and diversity of the diffusion model to generate more complex and creative animations. Besides them, extending InsActor to accommodate different human body shapes and morphologies is also an interesting direction.

From a societal perspective, the application of InsActor may lead to ethical concerns related to how it might be used. For instance, InsActor could be exploited to create deceptive or harmful content. This underscores the importance of using InsActor responsibly.

## Acknowledgment

This research is supported by the National Research Foundation, Singapore under its AI Singapore Programme (AISG Award No: AISG2-PhD-2021-08-018), NTU NAP, MOE AcRF Tier 2 (T2EP20221-0012), and under the RIE2020 Industry Alignment Fund - Industry Collaboration Projects (IAF-ICP) Funding Initiative, as well as cash and in-kind contribution from the industry partner(s).

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
