# OpenReview forum: "InsActor: Instruction-driven Physics-based Characters"
_NeurIPS.cc/2023/Conference — NeurIPS 2023 poster_

### Official Review · Reviewer_7eMQ · 2023-06-30

**Soundness:** 3 good
**Presentation:** 4 excellent
**Contribution:** 2 fair
**Rating:** 7
**Confidence:** 2

**Summary:**

This work proposes InsActor, a framework for instruction-driven character animation. Given human instruction and/or way points, InsActor first leverages a diffusion model to generate state sequences of the character. Next, it uses a skill embedding model to convert the state sequence into physically plausible trajectories with state and actions.

**Strengths:**

1. The paper presents the motivations, setups, and methods quite well. The diffusion model for state sequence generation and conversion from state sequence to physical trajectories are described in sufficient details.

2. The paper provides sufficient experiments and comparisons to show it out-performs the baseline method. The hierarchical design is also justified with ablation studies.

**Weaknesses:**

1. The works appears to have limited novelty, as it is somewhat a straightforward combination of character motion synthesis with diffusion methods such as [13] and a low-level trajectory optimization framework.

2. This work seems to miss a comparison with a straightforward method for the low-level policy, since the second stage can be formulated with a standard trajectory optimization problem. Since Brax is also a differentiable simulator, trajectory optimization should be easy to set up and in principle, it can be used to obtain an optimal solution in terms of distances between the states generated from the first stage and states from a physically plausible trajectory. I think this work could include results and analysis from trajectory optimization or explain why it is not feasible here.

**Questions:**

Suggestion on writing:
- line 92: I believe angular velocity $q$ of a link is not the derivative of rotation, and the symbol $q$ clashes with the notation of joint position $q$. This description needs to be clarified to avoid confusion.

**Limitations:**

The paper has provided adequate discussion of limitation.

---

> ### Author Rebuttal · Authors · 2023-08-10
>
> We thank the reviewer for the thoughtful comments. We would like to address your concern as follows:
>
> > Q1: The work appears to have limited novelty, as it is somewhat a straightforward combination of character motion synthesis with diffusion methods such as [13] and a low-level trajectory optimization framework.
>
> This is a great point for us to clarify! We would like to highlight that this paper is more focused on studying a scalable framework for the new task of text-to-motion generation, instead of any individual component. We have conducted rigorous evaluation and fair comparison on various different design choices using state-the-art motion planning algorithm (MotionDiffuse) and a state-of-the-art tracking algorithm (DiffMimic) on a largest text-to-motion dataset available to identify the best framework, in which we believe is our core contribution in this work. Also, we believe that our framework will benefit from a better motion generation model and motion tracker in the future.
>
> There are two main points we would like to highlight of InsActor:
>
> 1. Scalability: Building a scalable system to achieve human instructed animation generation is highly non-trivial. Compared with previous works, InsActor understands more general human language in terms of the motion database size, where most related work PADL works on 131 individual clips, for a total of approximately 9 minutes, and InsActor works on HumanML3D dataset with  14,616 motions for a total of 28.59 hours (before filtering out physically impossible motions like sit on a chair). Training a general low-level policy handling various motions is extremely difficult as it may take up to hours to train a low-level policy to perform one single motion, e.g. backflip. InsActor is the first system to work on such a large-scale amount of data.
>
> 2. Thoroughness: As the first attempt for large scale physically-aware language guided animation generation, InsActor evaluates and benchmark various alternatives to gain insight to better push the task forward. With extensive experiments, we show that directly outputting actions and state sequences like Diffuser[13] or Decision Diffuser[1] using state of the art generative models may fail in this task, which requires accurate continuous control. Instead, a proper decomposition of the task should be performed in order to derive an efficient solution.
>
> Thus, InsActor is not merely a combination but an important baseline for future work as also acknowledged by reviewer s1C4. InsActor is scalable as the data scale grows and provides insights to the problem of language guided animation generation.
>
>
> > Q2: This work seems to miss a comparison with a straightforward method for the low-level policy, since the second stage can be formulated with a standard trajectory optimization problem. Since Brax is also a differentiable simulator, trajectory optimization should be easy to set up and in principle, it can be used to obtain an optimal solution in terms of distances between the states generated from the first stage and states from a physically plausible trajectory. I think this work could include results and analysis from trajectory optimization or explain why it is not feasible here.
>
> We chose not to use trajectory optimization for several reasons:
>
> 1. Computational Efficiency: While employing standard trajectory optimization alongside the differentiable simulator is possible, optimizing trajectories for long motion sequences can be time-consuming. Each generated trajectory necessitates optimization, leading to potential delays. In contrast, the utilization of a trained policy enables fast inference. Given the emphasis on engaging with human users within our work's scope, we chose not to adopt trajectory optimization due to computational efficiency.
>
> 2. Robustness: The application of trajectory optimization methods such as [A] typically yields a single trajectory. However, our envisioned use cases for InsActor involve animations and gaming scenarios. In these dynamic environments, robustness is important. Inherent noise in state estimation or random perturbations like pushing in the environment affecting characters are pervasive. Consequently, the efficacy and robustness of the low level policy becomes pivotal in ensuring InsActor's robustness during real-world deployment. Simple trajectory optimization is unable to handle such cases. In addition, we attach the tracking error of the learned low-level policy in the PDF file, which is low and is on par with previous motion tracking works.
>
> 3. Flexibility: A learned policy can adapt to various situations and environments. Once trained, a policy can generalize its behavior to new situations that it hasn't explicitly encountered during training. This is important for future development of InsActor like enabling object interaction and accommodating more intricate animations. The current framework of InsActor enables seamless integration of object interaction into the framework with the low level tracking module.
>
> In summary, the design is mainly due to the consideration of the application scenario and use cases. Nevertheless, we agree with the reviewer that including such comparison would make the experiment results more thorough and we will add them in the final version due to time limit during the rebuttal period.
>
> Additional References:
>
> [A] Gärtner et al., “Differentiable Dynamics for Articulated 3d Human Motion Reconstruction”, CVPR 2022.

---

> > ### Comment · Reviewer_7eMQ · 2023-08-21
> > **Response to rebuttal**
> >
> > I would like to thank authors for the explanation. I agree that the main contribution of the work is a scalable pipeline for character motion synthesis, and I would keep my original **accept** recommendation.
> > However, after discussion with other reviewers, I think the quality of character motion could still be improved (penetrating and jittering artifacts). Without improving the quality, I do not think computational efficiency or flexibility need to be prioritized. Therefore, I still encourage including trajectory optimization (potentially adding stronger contact constraints to reduce artifacts) to see if it would help improve the synthesis quality.

---

### Official Review · Reviewer_s1C4 · 2023-07-03

**Soundness:** 2 fair
**Presentation:** 3 good
**Contribution:** 3 good
**Rating:** 5
**Confidence:** 4

**Summary:**

This paper tackles the problem of generating text-conditioned character animation that is physics-based. It proposes a two-stage approach that first generates a kinematic motion conditioned on text using diffusion, and then tracks this motion with a physics-based motion VAE in the learned latent space. Experiments on the common KIT-ML and HumanML3D benchmarks show improved performance over prior work in physics-based motion from text, and show the ability to specify goal waypoints for the motion to hit while following human text prompts.

**Strengths:**

Physics-driven text-to-motion is an important problem and conditioning on free-form text input has not really been tackled in the literature, so the paper is novel in that respect. Diffusion has shown promising results recently, but these are all kinematic.

The proposed idea of using high-level diffusion followed by physics-based tracking is simple and solid. It would make a good first baseline for future work in this area.

Technically, InsActor uses a physics-based motion VAE to track kinematic motion which is novel, and it uses differentiable physics to train rather than a learned world model or RL. If this approach really does work to track general motions in the HumanML3D dataset, it would contribute an alternative to recent RL approaches that can be difficult to generalize.

The proposed method is evaluated on HumanML3D and KIT, which are the most relevant benchmarks for the text-to-motion task. Also, the DReCon baseline in Tables 1 and 2 is an important baseline that uses the alternative state-based approach to tracking.

The supplementary video and figures are visually pleasing, and I appreciate that the supp video is extensive and shows many results. The shown demo is also cool, and demonstrates fast generation capabilities (relative to other diffusion approaches).

**Weaknesses:**

To better motivate the need for physics in text-to-motion, there should be a comparison between the proposed InsActor and a state-of-the-art kinematic diffusion model like MDM [33] added to Tables 1 and 2. Currently some numbers in these tables, e.g. FID and diversity, are worse than those reported in MDM, and it’s not clear why that’s the case since the high-level planner in InsActor is very similar. Does this high-level planner perform worse than MDM? Or does the physics-based motion tracking somehow have a large effect on motion quality and diversity? The high-level policy ablation reported in Table 4 takes a step in this direction, but it is trained on rollouts from the motion VAE instead of directly on mocap data as done in MDM and other text-to-motion diffusion models.

Looking at video results at 0:46 and 01:54, I don’t think the high-level diffusion planner is on par with recent models like MDM. Both with and without waypoint guidance there are some significant artifacts like jittering and skating, some of which seem to be affecting the final motion from InsActor (e.g. some noisy popping of limbs and unnatural sliding). I understand this planner is not necessarily the main contribution, but I think poor kinematic motions from the planner undermines the comparison to the target-state tracking policy DReCon, which may perform better when operating on, e.g., outputs from MDM that better reflect realistic motion. Since the low-level motion VAE model for tracking (Sec 4.2) is a key contribution of the work, it’s very important to justify that it is necessary by showing that the DReCon baseline is still inferior when operating on more reasonable kinematic inputs.

The methods Sec 4 is missing some details that could improve understanding and reproducibility:
* The tasks states described in L92 are all local, so how is the global root trajectory modeled in the diffusion Sec 4.1?
* L154: if the pose state is in the local frame, how is inpainting performed to ensure motion meets a global target waypoint?
* What is the architecture of the diffusion model? Is it using 1D convolutions as in Diffuser or a transformer as in other human motion diffusion models? $\mu$ and $\Sigma$ in Eqn 2 are never defined. In general, I’m wondering why not use a SOTA motion diffusion model out-of-the-box for this high-level planning component?
* L185: is the low-level motion VAE trained directly on outputs of the diffusion model or on mocap data from the dataset? If on mocap, why is the encoder (Eqn 4) expected to produce reasonable results when operating on noisy and unrealistic pose transitions?
* Similarly, Sec 5.3 shows robustness to perturbations from boxes, but is this kind of perturbation seen in training of the low-level policy too? If not, how does this robustness arise without using RL for training (i.e. without some exploration).

An evaluation on the low-level tracking component by itself would be very helpful. E.g. reporting tracking errors for the latent policy from InsActor compared to DReCon for both motion-captured and diffusion-generated motions. The current metrics in Tables 1 and 2 were designed for kinematic text-to-motion models, and I would think are mostly influenced by the diffusion planner which is the same for InsActor and DReCon, so a tracking-only evaluation could help parse the difference in performance. There are also open-source RL physics-based trackers that may be worth considering, e.g. [Luo et al., Dynamics-Regulated Kinematic Policy for Egocentric Pose Estimation, NeurIPS 2021].

**Questions:**

Overall, I think a general physics-based text-to-motion model is an important and novel direction, and the hierarchical approach of diffusion planning with physics-based tracker could be a strong baseline going forward. But I’m mainly concerned that the quality of the output from the high-level diffusion planner is compromising the comparison to DReCon and therefore the need for a latent motion VAE tracker has not been fully justified. I would really like to see how InsActor performs when plugging in a SOTA diffusion model like MDM [33] as the planner. Moreover, an evaluation of standalone tracking performance would make the comparison between the two tracking approaches (latent vs state-based) much more clear.

Some other comments and suggestions that didn’t have an influence on my rating:
* The related work (Sec 2) is missing relevant physics-based human animation methods and a discussion of why they are difficult to scale up to the general text-to-motion task. E.g. [Peng et al., ASE: Large-Scale Reusable Adversarial Skill Embeddings for Physically Simulated Characters, SIGGRAPH 2022] [Won et al., A Scalable Approach to Control Diverse Behaviors for Physically Simulated Characters, SIGGRAPH 2020], etc..
* Concurrent work PhysDiff [Yuan et al., arxiv 2022] is an alternative approach to adding physicality to text-to-motion diffusion and could be discussed in future revisions of the paper. Also related, concurrent work Trace and Pace [Rempe et al., CVPR 2023] gives controllability over physics based characters with guidance of a diffusion planner.

===================== After Rebuttal ============================

After considering other reviews and discussions with authors and between reviewers, I have decided to slightly raise my score and am leaning towards accept. I think the paper lays out a compelling kinematic diffusion + physics-based tracking idea that can serve as a baseline and inspire improvement in each component of the system including differentiable simulation, motion diffusion, and physics-based tracking. The evaluations show that this hierarchical approach works better than state+action diffusion, and that tracking with a latent model is more robust than direct state-based tracking for planned motions from MotionDiffuse.

However, I am still quite concerned about the qualitative results and would really encourage the authors to update the paper text to discuss these qualitative issues such that future work can pursue important directions (e.g., the choice of Brax and differentiable simulation in general rather than RL, and the noisy plans from MotionDiffuse especially in the waypoint setting). It would also be good to clarify why in Table 2 of the rebuttal doc, motion quality (FID) drops significantly from kinematic “Planner” (b) output to full physics-based InsActor (c), indicating that adding physics-based tracking is not necessarily improving motion realism despite it being physically constrained (unlike in PhysDiff). I encourage the authors to show some video results of the planner output vs InsActor on regular text-to-motion (not the waypoint setting) to demonstrate the difference and potential advantages/disadvantages of using the latent tracking technique vs RL and more reliable simulators as in DReCon.

==========================================================

**Limitations:**

Limitations are sufficiently discussed in Sec 6.

---

> ### Author Rebuttal · Authors · 2023-08-10
>
> We thank the reviewer for the detailed comments. Due to the character limit, we would like to address your major concerns here. For more detailed questions (e.g. tracking details), we are glad to discuss them during the discussion phase.
>
> > Q1:  To better motivate the need for physics in text-to-motion, there should be a comparison between the proposed InsActor and a state-of-the-art kinematic diffusion model like MDM [33] added to Tables 1 and 2. Why FID and diversity are worse than those reported in MDM? The high-level policy ablation reported in Table 4 takes a step in this direction, but it is trained on rollouts from the motion VAE instead of directly on mocap data as done in MDM and other text-to-motion diffusion models.
>
> This is a great point for us to clarify! Unfortunately, since MDM has a different output domain than our state trajectories, we can not directly compare MDM in Table 1/2. However, we would like to clarify that our high-level planner is adapted from MotionDiffuse[A], a state of the art text-to-motion model that is on par with MDM. Quantitatively, Table 2 in the appended PDF shows that our planner achieves strong generation performance.
>
> As we report in the general response, there is indeed a gap between the kinematic and physically simulated setting, which demonstrates the challenge of this new task. The gap is led by the fact that the kinematic motion generator is not physically-plausible, which can not be reflected on the previous evaluation metrics like FID. This issue has also been concurrently observed in a recent work PhysDiff[B], which proposed a trajectory optimization approach to fix the generated motion. Different from PhysDiff which is in the kinematic setting, our character moves in a simulated environment and the physical plausibility is more rigorously ensured. In addition, motion generation models tend to produce sequences that appear appealing but are physically implausible. In the process of grounding the state sequence in physics, certain compromises are made on visual aesthetics in order to adhere to physical realities.
>
> The "high-level" ablation study in Table 4 refers to the framework proposed in the Diffuser[13], where a generative model predicts both state sequence and action sequence. This study is to show the importance of our low-level skill embedding, and does not reflect the capability of our high level planner.
>
> > Q2: Looking at video results at 0:46 and 01:54, I don’t think the high-level diffusion planner is on par with recent models like MDM. Since the low-level motion VAE model for tracking (Sec 4.2) is a key contribution of the work, it’s very important to justify that it is necessary by showing that the DReCon baseline is still inferior when operating on more reasonable kinematic inputs.
>
> Qualitatively, we do notice that our generated plans have more jittering than motions generated by either MDM or MotionDiffuse. This could be caused by the fact that MotionDiffuse uses temporal smoothing in the visualization but we did not smooth our plans in our visualization. However, we show in Table 1 in the appended PDF that plan smoothing has a minimal effect on the tracking result.
>
> In terms of the concern that a poor kinematic motion undermines the comparison to DReCon, we would like to highlight that making an executable plan can be very difficult for generative planners. This has also been discussed by a recent work PhysDiff[B], which shows that the physics plausibility generated by MDM has a large room for improvement. Particularly, we observe that the invalid state transition increases when under the waypoint conditioned setting. The mismatch between global motion and locomotion makes it extremely difficult for DReCon-like trackers to track. Note that DReCon has only been verified on a small scale 10-minute motion database with high quality motion data generated by Motion Matching.
>
>
> > Q3: The methods Sec 4 is missing some details that could improve understanding and reproducibility: An evaluation on the low-level tracking component by itself would be very helpful. There are also open-source RL physics-based trackers that may be worth considering, e.g. [Luo et al., Dynamics-Regulated Kinematic Policy for Egocentric Pose Estimation, NeurIPS 2021].
>
> We thank the reviewer for pointing it out. Due to the character limit, we will address your main concern by providing an evaluation of the tracking module and will add more details regarding the tracking module in future iterations or the discussion period.
>
> As shown in Figure 1 in the appended PDF, our motion tracker achieves excellent tracking performance with a pose error smaller than 0.05m. We also show in Table 2 in the appended PDF that our implemented DReCon trackers achieves 0.086 FID tracking the test dataset. Although motion tracking is not our main contribution, we highlight that our low-level policy is trained on a very large motion database and achieving the tracking performance is challenging.
>
> Thanks for bringing the work by Luo et al. to our attention! However, although the work provides an RL-based motion tracker, its setting is a simplified version of ours. Concretely, in their setting, there is a residual force, i.e., hand of god, which is an external force acted on the character. Although the low-level control can be simplified and the motion quality can be improved with additional external force (hand of god), the generated motion is not technically physically plausible due to the invisible force. On the other hand, in our setting,  all forces are produced by the character itself, which follows the standard physics-based character animation setting and hence is more challenging.
>
> Additional References:
>
> [A] Zhang et al., “MotionDiffuse: Text-Driven Human Motion Generation with Diffusion Model”, ArXiv, 2022.
>
> [B] Yuan et al., “PhysDiff: Physics-Guided Human Motion Diffusion Model”, ICCV, 2023.

---

> > ### Comment · Reviewer_s1C4 · 2023-08-14
> > **Followup question**
> >
> > I would like to thank the authors for their thorough response to my points and questions.
> >
> > Could you please expand on why the results from MDM/MotionDiffuse (in Table 1 of the attached document) are not directly comparable to the output of "Planner" in Table 2? Both methods output a set of joint positions/rotations that can be used to compute the metrics, correct? Is it because InsActor use a different skeleton topology than the SMPL body used in MDM?
> >
> > Thanks!

---

> > > ### Author Response · Authors · 2023-08-15
> > > **Answer to followup question**
> > >
> > > Thank you for your question!
> > >
> > > Yes, the difference in topology is part of the reason. Additionally, variations in body proportions and motion representation prevent us from directly adopting the evaluation code from text-to-motion works.
> > >
> > > **Topology Difference:** Our character has a topology that slightly differs from a standard SMPL model, as it lacks the middle joint of the spine, referred to as "spine 1."
> > >
> > > **Body Proportion Difference:** The character we used originates from DeepMimic [23]. Its body proportion differs from the template skeleton employed in the standard evaluation code. This leads to variations in forward/inverse kinematics results.
> > >
> > > **Motion Representation Difference:** InsActor utilizes a standard state representation in physics-based character animation. This includes link position, rotation, linear velocity, and angular velocity in the global frame. It's worth noting that a link refers to a rigid segment of an articulated body, like arms and legs, connected by joints. Conversely, motion representation in text-to-motion works relies on joint information in the local frame and excludes per-joint angular velocity.
> > >
> > > Due to these differences in generation space, we constructed our evaluation pipeline from scratch, following the guidelines of Guo et al. [7]. This process included training the contrastive models instead of leveraging pretrained ones. As a result, the numerical outcomes aren't directly comparable.
> > >
> > > Furthermore, converting from the InsActor representation to the text-to-motion representation is feasible. However, such conversions come with certain losses — for instance, when changing the InsActor link global positions to the SMPL joint local positions. Nonetheless, we acknowledge that employing the text-to-motion codebase to assess the converted InsActor output offers a direct comparison with purely kinematic baselines and might yield valuable insights. We will include this converted comparison in a future version.

---

> > > > ### Author Response · Authors · 2023-08-20
> > > > **Followup**
> > > >
> > > > We hope that our response clarifies why the numbers in the two tables are not comparable.
> > > >
> > > > In our rebuttal,
> > > > - We have clarified that our planner is adapted from a state-of-the-art text-to-motion approach and achieves a strong performance on kinematic motion generation.
> > > > - The visualization of the planner contains jittering because we did not apply smoothing, and smoothing has a minimal effect on the tracking results.
> > > > - Our low-level controller achieves excellent tracking performance with a pose error of less than 0.05. The motion tracker implemented for DReCon can successfully track real motions with an FID of 0.086.
> > > >
> > > > As the discussion phase is about to close, please kindly let us know if there is anything to be further clarified.

---

### Official Review · Reviewer_K1zb · 2023-07-04

**Soundness:** 3 good
**Presentation:** 2 fair
**Contribution:** 2 fair
**Rating:** 5
**Confidence:** 5

**Summary:**

This work proposes InsActor, a physics-based character control framework that enables controlling agents to follow high-level instructions. It employs a high-level diffusion motion planner and a low-level character controller to achieve mapping between text-based instructions and physics-based motion control. By creating waypoints and desired state transitions, the diffusion motion planner specifies the motion plan that the low-level motion controller follows. The low-level motion controller consists of a pre-trained VAE encoder to translate diffusion state into actionable latents for control.

**Strengths:**

- I find the idea of using a higher-level motion diffusion model to specify waypoints and states, and then using a lower-level motion controller to follow them, intuitive and easy to understand. This formulation is flexible and creates the ability to have both high-level control of motion through text and slightly lower-level control through waypoints.
- The low-level skill discovery module based on diffMimic itself seems like a significant contribution, as it has the potential to be used in many tasks and enables character control. Using a VAE to translate invalid state transitions into actionable latent codes for stable control has great potential in other downstream tasks.
- Experiments show that the methods outperform the in-house implementation of the state-of-the-art methods PADL and DReCon.

**Weaknesses:**

- The main weakness I find in this work is its qualitative results. The simulated character is jittery, floaty, and appears to have foot sliding, which is unexpected for a physics-based method. For instance, at the 1:43 mark of the video, InsActor's feet quickly shuffle in a way that should be impossible in a physics simulator. The character seems overall drunk when walking, and there are occasional high-frequency jitters in the root (or camera?), for instance, at 00:48. I am not sure what could have caused this issue: is it the policy? or the setting/fidelity of the Brax simulator?
- Similar to the previous point, the visual results of the implemented PADL and DReCon are far inferior to their original shown results. The movement is unnatural and jittery. I understand that there is no official implementation provided, but similar scenarios could be recreated in InsActor to create visual comparisons. PADL's generated motion is stable and physically realistic, unlike the ones shown in the provided video. As motion generation is largely visual, the provided quantitative results do not really provide that much insight into the real performance of the method.
- Visualization should be provided for the high level diffusion planer. Since states are directly generated by the planner, they should be visualized and compared with existing models such as motion diffusion model (MDM). The lower level controller, combined with the latent skill decoder $p_\psi$, forms an imitator that follows specified states. How well does the diffusion model generates the state and how “invalid” are they?
- The claim of "long, unstructured human commands" is overstated. The tested text instructions are still short, clear, and close to the ones in the dataset.
- There are many missing details about the performance of the lower-level skill discovery module. If the whole HumanML3D dataset is used for training, how well can the skill embedding encapsulate the motion described in the dataset?

**Questions:**

My main concern is the fidelity of the generated motion. Why does the character appear to be floaty and jittery? Is it the learned policy or the simulator settings?

---
After rebuttal, my main concern about the fidelity of the generated motion returned. However, after discussion with other reviewers and the authors, I believe that the this issue could be a problem of the underlying simulator and should not undermine the overall contribution of the framework. Thus, I raise my score to borderline accept.

---

**Limitations:**

Limitations are addressed.

---

> ### Author Rebuttal · Authors · 2023-08-10
>
> We thank the reviewer for helpful comments. We would like to address your concern as follows:
>
> > Q1: The main weakness I find in this work is its qualitative results. The simulated character is jittery, floaty, and appears to have foot sliding, which is unexpected for a physics-based method.
>
> Our  results are directly exported from the Brax simulation engine, which conform to the physic-based character animation settings. However, we do observe some artifacts in our qualitative results. They may come from multiple sources:
>
> 1. Imperfect motion retargeting. For the main results in the paper and the teaser video, we retarget the simulated motion to a character (a white Y-Bot from Adobe Mixamo) for better visual appearance. However, to preserve the motion content, we copy the rotation angles and add a vertical offset, which can not fully address the difference in bone lengths and introduces additional physics implausibility like foot floating and foot penetration.
> 2. Brax simulator smoothens the contact to allow differentiability, which is a known issue [28]. This approximation could lead to possible foot sliding or wrong collision. Brax is relatively new and differentiable physics simulation is still in rapid development, therefore we believe that the issue can be addressed by a better DPS.
> 3. Imperfect motion planning and motion tracking policy. Although we are using a state-of-the-art text-to-motion diffusion generator (MotionDiffuse), the generated motion plan can contain jittery and impossible state transitions. For motion tracking, although we used the state-of-the-art motion tracking algorithm (DiffMimic), tracking a very large motion database is non-trivial and could lead to performance degradation.
>
> > Q2:  Similar to the previous point, the visual results of the implemented PADL and DReCon are far inferior to their original shown results.
>
> Our task is much more challenging than DReCon and PADL. DReCon tracks 10-minute motion captures, PADL tracks 9-minute captures, while our model uses HumanML3D, a dataset with 28.59 hours of diverse motions. This large dataset presents significant tracking challenges. Since neither PADL nor DReCon have official implementations, to compare fairly, we reproduce them using DiffMimic. Our tracker achieves low pose tracking error on motion datasets for high-quality reproduction. We refer the reviewer to Table 2 in the uploaded PDF file for more details.
>
> > Q3: Visualization should be provided for the high level diffusion planner.  How well does the diffusion model generate the state and how “invalid” are they?
>
> We have provided some plan visualizations in the supplementary videos 00:45 and 01:54. Quantitatively, Table 2 in the appended PDF shows that our planner achieves strong generation performance. Examples of "invalid" state transitions are demonstrated in our supplementary video at 00:45, revealing artifacts like floating, foot sliding, and jittering. These issues even signify in the waypoint heading task as demonstrated in our supplementary video at 01:54. As a result, directly tracking these plans can easily lead to failure, as shown in the alongside DReCon results, whereas InsActor overcomes this with skill embeddings. Notably, physics-plausibility in motion generation, addressed in PhysDiff[A], aligns with our findings as MDM lacks it. Incorporating physics-based optimization enhances MDM's scores on physical plausibility, supporting our observation of infeasible state transitions.
>
> > Q4: The claim of "long, unstructured human commands" is overstated. The tested text instructions are still short, clear, and close to the ones in the dataset.
>
> Our claim regarding previous works' inability to handle "long, unstructured human commands" refers to longer and more complex language we address. For example, PADL processes sentences like "jump and swing sword down" and "shield charge forward," using Multiple Choice Questions to structure input. In contrast, InsActor directly interprets intricate language, such as "the person raises their left foot up to their knee and then kicks their right foot out, then returns foot to their knee," as demonstrated in our supplementary video. Additionally, InsActor comprehends more general human language due to the larger motion database size. While PADL operates on 131 individual clips totaling around 9 minutes, InsActor employs the HumanML3D dataset with 14,616 motions spanning 28.59 hours.
>
> We acknowledge potential misunderstanding arising from "long, unstructured human commands" phrasing and will address this in our next version. Nevertheless, leveraging our extensive text-motion dataset, we anticipate our language understanding capability to improve with more descriptive datasets in the future.
>
> > Q5: There are many missing details about the performance of the lower-level skill discovery module. If the whole HumanML3D dataset is used for training, how well can the skill embedding encapsulate the motion described in the dataset?
>
> In the general response, we show that the tracking error is low and is on par with previous motion tracking works. In addition, the FID is 0.086 when tracking ground truth motions. We agree that using a single policy network to encapsulate all motion skills on a large motion dataset can be suboptimal. As previous works in motion tracking have shown[B], an MoE ensemble can largely improve the model capacity and better capture dynamic moves like break dancing. However, adopting the system to skill embedding is non-trivial and is out of the scope of this paper that focuses on understanding human language instruction. Nonetheless, we believe that a stronger skill embedding module is an important next step and will be a crucial addition to our framework.
>
> Additional References:
>
> [A] Yuan et al., “PhysDiff: Physics-Guided Human Motion Diffusion Model”, ICCV 2023
>
> [B] Won et al., “A Scalable Approach to Control Diverse Behaviors for Physically Simulated Characters”, SIGGRAPH 2022

---

> > ### Comment · Reviewer_K1zb · 2023-08-16
> > **Reviewer Response**
> >
> > I thank the authors for the detailed response!
> >
> > My concern about the motion quality remains. I understand the artifacts such as penetration, foot sliding, or wrong collision could be a result of Brax simulator, but the ones shown in the videos are very obvious and jarring. At a bare minimal the collision and penetration needs to properly modeled to ensure physically plausibility, otherwise what is the purpose of using a simulator? Claiming that the motion is physically plausible while the generated motion has large non-physical artifacts seems disingenuous. If the physics simulator (Brax) is the issue, then more investigation needs to be made or a different simulator used. I understand that DPS is under rapid development, but enjoying its benefits (being differentiable) while not considering/discussing its drawbacks seems to be misleading for the community.
> >
> > Purely diffusion-based methods for motion generation seems to be generating much more smooth and good-looking motion overall. Results shown from the MotionDiffuse paper seems to be better than the result shown here (00:45 and 01:54). What could be the cause?
> >
> > For the tracking error, I think acceleration error and velocity error, in addition to a MPJPE is needed for better a better picture. How many sequences can be tracked successfully?

---

> > > ### Author Response · Authors · 2023-08-17
> > > **Answer to Reviewer Response**
> > >
> > > Thanks for opening up about your concerns!
> > >
> > > >  Claiming that the motion is physically plausible while the generated motion has large non-physical artifacts seems disingenuous.
> > >
> > > All of our results are fully simulated in a well-received physics engine Brax, so we respectfully disagree that claiming the motion is physically plausible is disingenuous. It is noteworthy that all physics simulation engines like Bullet and IssacGyms do not guarantee perfect physics simulation, which does not undermine the significance of the research built on top of them.
> > >
> > >
> > > >  If the physics simulator (Brax) is the issue, then more investigation needs to be made or a different simulator used.
> > >
> > > We agree that more alternatives to Brax can be explored which may lead to better visual results. However, the training efficiency of DiffMimic, which is benefited from the differentiability of Brax, is key to conducting the research on a large-scale language-to-motion dataset. Note that as described in ScaDiver, training a tracker with a non-differentiable physics simulator like PyBullet on the AMASS dataset takes tremendous efforts. Training a single expert policy takes up to 6 days, and training the Mixture-of-Expert controller takes another 10 days. Despite the possibility that alternative physics simulators can reduce the artifact, Brax is a reasonable choice for us at this point considering that it has a significantly better training efficiency and our research focus is not on motion tracking.
> > >
> > > >  I understand the artifacts such as penetration, foot sliding, or wrong collision could be a result of the Brax simulator, but the ones shown in the videos are very obvious and jarring.
> > >
> > > As stated in the rebuttal, Brax is not the only source for visual artifacts. Foot penetration/floating is resulted of the naive motion retargeting from the simulated character to the visualization character rendered in __Blender__. The jittering is a result of imperfect motion plans. Brax is mainly responsible for the foot sliding problem. We do not observe foot penetration/floating and jittering when tracking ground truth motions with Brax.
> > >
> > > > At a bare minimal the collision and penetration needs to properly modeled to ensure physically plausibility, otherwise what is the purpose of using a simulator?
> > >
> > > Physics simulation also enables interactiveness with the physical surroundings, which purely kinematic generation inherently falls short of.
> > >
> > >
> > > > I understand that DPS is under rapid development, but enjoying its benefits (being differentiable) while not considering/discussing its drawbacks seems to be misleading for the community.
> > >
> > > We agree that more limitations of using DPS should be discussed in the paper. We will improve on that in future versions.
> > >
> > > > Purely diffusion-based methods for motion generation seems to be generating much more smooth and good-looking motion overall. Results shown from the MotionDiffuse paper seems to be better than the result shown here (00:45 and 01:54). What could be the cause?
> > >
> > > As explained in the rebuttal, MotionDiffuse uses an additional smoothing in the visualization: *This could be caused by the fact that MotionDiffuse uses temporal smoothing in the visualization but we did not smooth our plans in our visualization.*
> > >
> > > Specifically, the temporal smoothing function is used at text2motion/tools/visualization.py#L22 in their official repository. Therefore, the visualization in MotionDiffuse does not have jittering. Note that this smoothing module is __not__ applied in the evaluation of MotionDiffuse. Following the same protocol, we did not apply the smoothing module in our experiments. Please see the attached PDF on how adding this smoothing module will not affect the final simulated results.
> > >
> > > > For the tracking error, I think acceleration error and velocity error, in addition to a MPJPE is needed for better a better picture. How many sequences can be tracked successfully?
> > >
> > > The definition of success in tracking can be vague. The fall rate successfully converges to zero after few hours of training, where falling is defined as the pelvis dropping below 0.2 meter. In particular, we would like to refer the reviewer to the attached PDF on how DReCon achieves a very low FID using the tracking module when tracking the HumanML test set. From this perspective, the tracking module can track most sequences successfully as indicated by the FID.
> > >
> > > Thanks for the suggestion of MPJPE, acceleration error, and velocity error. We are aware that they are common metrics in human pose estimation, where the latter two penalize unnatural jitterings. We will consider additionally including these metrics in justification of our tracking performance.
> > >
> > > Given the limited time and restricted rebuttal policy, we are unable to provide additional demo videos/visualization or experiments to fix the issue. We thank you for your valuable questions and will further improve them in our revisions

---

> > > > ### Comment · Reviewer_K1zb · 2023-08-18
> > > > **Reviewer Response followup**
> > > >
> > > > I thank the authors for the prompt response.
> > > >
> > > > > We do not observe foot penetration/floating and jittering when tracking ground truth motions with Brax.
> > > >
> > > > I am not referring to the Blender render but the Brax simulation result, as rendered by the capsule humanoid.
> > > >
> > > > At 01:25, InsActor in **Brax** is clearly stepping into the ground on the right foot. Same as 01:27.
> > > >
> > > > At 01:39, InsActor is jittering up and down, same at 01:52. The humanoid is either bouncing up and down or floating.
> > > >
> > > > At 02:11, the humanoid seems to be gliding on ice.
> > > >
> > > > The ablation in 02:50, all three seeds has large quick feet shuffling and jitter.
> > > >
> > > > I understand no physics simulation is perfect, real-world physics, but the result shown in the supplementary video is quite below the SOTA quality expected from physics-based methods.
> > > >
> > > > > The definition of success in tracking can be vague. The fall rate successfully converges to zero after few hours of training, where falling is defined as the pelvis dropping below 0.2 meter.
> > > >
> > > > I am glad to see that fall-rate converges to zero, though an average MPJPE and physics-based metrics would still be helpful to gauge the tracker performance.

---

> > > > > ### Author Response · Authors · 2023-08-20
> > > > > **Answer to Reviewer Response followup**
> > > > >
> > > > > We sincerely appreciate your clarifications. We agree that the mentioned artifacts are potentially caused by simulation. Unfortunately, the only way to alleviate this issue is to improve the general physics simulation quality of Brax, which we believe is out of the scope of this work at this point. Theoretically, the proposed InsActor is agnostic to the backbone physics engine. With an improved DPS featuring refined contact approximations, InsActor would undoubtedly yield better visual outcomes. We defer this exploration to future investigations.

---

### Official Review · Reviewer_Mekg · 2023-07-07

**Soundness:** 3 good
**Presentation:** 3 good
**Contribution:** 3 good
**Rating:** 7
**Confidence:** 4

**Summary:**

The paper proposes an approach for generating physically plausible human motion from open text prompts. The approach combines high-level trajectory generation with guided diffusion and a low-level skill model encoded in a latent space to correct for physical plausibility during execution in simulation with PD control. The proposed approach is compared against existing motion generation approaches with a set of quantitative metrics and qualitative evaluation.

**Strengths:**

Overall this work soundly presents an improvement to SoTA in the space of human motion generation from text prompts. The use of guided diffusion to produce high-level trajectories is not surprising, nor is its effectiveness when combined with low-level skill models as a method of correcting physical implausibilities and errors. The paper is well written and clearly describes the architecture, training procedure, and use of guidance for waypoint following. Evaluation is sound and clearly demonstrates the strengths of the approach. I appreciate the use of multiple quantitative metrics and stochastic evaluation of many samples. This work benefits from the existence and availability of its component parts (Brax, ControlVAE, motion diffusion models), but connects them to achieve fairly good results. I’m always happy to see motion generation connected to physics simulation instead of remaining kinematic. The results of this work seem fairly robust to perturbation, which is a good signal for the usefulness of future iterations of the approach.


**Weaknesses:**

While the resulting motion is fairly good, it isn’t yet achieving the level of fidelity to make it generally useful in generative contexts (e.g. game or animation characters).
This work achieves its success from connecting other existing techniques, though the overall system is a sound contribution.
The stack is trained considering only the character’s joint space and a specific character’s physical constraints.


**Questions:**

Should cite Trace and Pace: Controllable Pedestrian Animation via Guided Trajectory Diffusion for similar architecture (guided-diffusion trajectories + low-level physical controller) https://arxiv.org/abs/2304.01893
Table 3 ordering of metrics is different from Metrics text section. Swap Multimodal Distance and FID in one or the other.
Is it feasible to generalize this stack to different body shapes, dynamics, to include objects or other kinds of waypoints?
Would it be possible/better to have some model-based or explicit low-level skill experts for specific sub-tasks (e.g. locomotion) to achieve higher fidelity in addition to general/open motion skills?


**Limitations:**

Discussion of limitations was pretty good and I appreciated the limitation notes throughout the paper. I did not see any mention of generalizing to other character morphologies (even different human body shapes). I currently believe this work is using a single embodiment as a strong assumption for achieving accuracy and physical plausibility. I would be interested to hearing more about the feasibility of generalization for applications.

---

> ### Author Rebuttal · Authors · 2023-08-10
>
> We thank the reviewer for constructive comments. We would like to address your concern as follows:
>
> > Q1: While the resulting motion is fairly good, it isn’t yet achieving the level of fidelity to make it generally useful in generative contexts (e.g. game or animation characters)
>
> This is a great point for us to elaborate more! As our approach delivers the first large-scale instruction driven physics-based character baseline, we also identify several aspects to improve the motion fidelity:
> 1. Simulation quality. Although all of our results are from physics simulation, the differentiable simulator may not generate high-fidelity motions. We note that the Brax engine has made approximations to achieve the differentiability. As a result, our resulting motion may have some artifacts as observed. This is a known issue in the state-of-the-art motion tracker DiffMimic that we use.  Nonetheless, we believe that this can be resolved by future efforts in developing a better differentiable physics engine.
> 2. Data Scale. Being a data-driven approach, we recognize that the data scale is an important bottleneck. Despite using the largest text-to-motion dataset HumanML3D, it is far smaller than billion-level text-to-image datasets. We believe that our approach serves as the first  scalable instruction-driven physics-based character animation baseline, and will continuously improve when more data is getting available.
>
> > Q2: This work achieves its success from connecting other existing techniques, though the overall system is a sound contribution. The stack is trained considering only the character’s joint space and a specific character’s physical constraints.
>
> Action generation in the joint space is the commonly adopted protocol in prior works[22,A] as they are general enough to achieve various kinds of motions. We use the same protocol in order to align with previous works.
>
> > Q3: Should cite Trace and Pace: Controllable Pedestrian Animation via Guided Trajectory Diffusion for similar architecture.
>
> Thanks for bringing this paper to our attention! This is an important related work for InsActor we shall discuss. We will add it into the manuscript in the final version.
>
> > Q4: Swap Multimodal Distance and FID in one or the other. Is it feasible to generalize this stack to different body shapes, dynamics, to include objects or other kinds of waypoints?
>
> We believe that our framework can be extended to include objects into the states. The high-level controller can generate motion plans conditioned on object states and the low-level controller will execute the interaction. For example, given a chair, a plan for a character to sit on the chair will be generated and executed without specifically training for the task. However, extending our current waypoint encoding system to encode object information including different types, sizes, and 3D locations is highly non-trivial. In addition, the current motion data scale for human-scene interactions may not match the complexity of the problem. Therefore, we do not consider object interactions in this work. Nonetheless, we do notice recent works like [A] that show promising results on physics-based character-object interaction, and view it as a vital future work. We believe our approach will serve as an important baseline that is extendable to character-scene joint modeling and expandable with additional human-scene motion data.
>
> We answer the question regarding different body shapes in Q6.
>
> > Q5: Would it be possible/better to have some model-based or explicit low-level skill experts for specific sub-tasks (e.g. locomotion) to achieve higher fidelity in addition to general/open motion skills?
>
> We agree that a single policy network for all motion skills on a large dataset isn't optimal. Previous motion tracking works have demonstrated that an MoE ensemble enhances model capacity and captures dynamic moves more effectively[B,C]. In these works,  the motion dataset is categorized into groups, and an expert policy is trained for each group. A higher-level policy can then determine the best expert policy deployment. Although expert ensemble is a promising approach to achieve high  motion fidelity, implementing an expert system from scratch requires large engineering efforts and could take a long time for training[C]. Since this work is less focused on motion tracking but more on language understanding, we choose DiffMimic which can be trained efficiently as shown in Figure 1 in the attached PDF.
>
> > Q6: I did not see any mention of generalizing to other character morphologies (even different human body shapes).
>
> 1. Different Morphology: This direction is intriguing but remains challenging due to current data limitations. Effective high-level planner training relies on ample text-motion pairs. However, data for non-human character morphologies is scarce, hindering high-level planner training. Cross-morphology transfer attempts[d] are intricate due to dataset motion diversity, constituting a significant independent contribution.
>
> 2. Varied Human Body: Body variation is an interesting topic and there are promising ways to achieve this level of generalization. For example, we can train the diffusion model with SMPL shape conditioning at the high level, and train a shape-invariant controller following [D]. However, as the first work to consider language-guided physical motion generation, our emphasis is on language understanding. We leave it for future study.
>
> We identify them as crucial future work and will add a discussion in future versions.
>
> Additional References:
>
> [A] Peng et al. “AMP: Adversarial Motion Priors for Stylized Physics-Based Character Control
> ” TOG 2021
>
> [B] Wagener et al. “MoCapAct: A Multi-Task Dataset for Simulated Humanoid Control.”NeurIPS 2022.
>
> [C] Won et al., “A Scalable Approach to Control Diverse Behaviors for Physically Simulated Characters”, SIGGRAPH 2022
>
> [D] Won et al., “Learning Body Shape Variation in Physics-based Characters”, TOG 2019

---

> > ### Comment · Reviewer_Mekg · 2023-08-21
> >
> > I want to thank the authors for their responses to my questions. The responses satisfied my queries and I remain positive about this work.

---

> > > ### Author Response · Authors · 2023-08-21
> > > **Thanks to Reviewer Mekg**
> > >
> > > We are glad to hear that our response addressed your concerns! Thank you for your recognition of our paper and your valuable suggestions.

---

### Official Review · Reviewer_ndRp · 2023-07-28

**Soundness:** 3 good
**Presentation:** 2 fair
**Contribution:** 3 good
**Rating:** 6
**Confidence:** 4

**Summary:**

The paper presents an approach for generating animations using a language-conditioned neural network in particular a diffusion model. The underlying goal is to generate physics based human character animation in an easy and intuitive fashion. The approach leverages a diffusion model to generate high-level states of the animated character which are then refined in a lower level module. This module is realized via an auto encoder which generates skill embedding representing the state transition in a compact fashion. A decoder is then used to synthesized the final action of the agent. The approach also uses an inpainting strategy to ensure that the agent goes through critical phases or points, i.e., this seems similar to the difference of keyframes and in-betweens in traditional animation. The paper compares the introduced approach to other recent methods on the topic, e.g., PADL and concludes among other things that more faithful animations (wrt to the original language query) are generated. The contributions of the paper is the combination of Diffusion and autoencoder models for motion generation, the inpainting strategy and waypoint system.

**Strengths:**

The paper addresses a really interesting and important problem that is very exciting. It is also one of the first papers to use diffusion models in this context and the results seem promising. The combination of in-painting with diffusion is also very clever (but details are missing). A BIG plus compared to other similar systems is that it does not require RL. For example, PADL requires multiple policies to be learned and later combined with each policy requiring 7 simulated years to be trained! That does not seem to be the case here since this is a purely supervised approach - even though substantial computation is required for all deep learning of course.
I really appreciated the use of contrastive models to evaluate the faithfulness of generated models. Diversity was also an interesting measure since in animation it is critical that the character shows a range of motions. Generally the paper is written in a way that can be followed (with caveats see below).

** Update after rebuttal **
I have increased my score to "weak accept". I appreciate the details provided by the authors - this helped better contextualize and understand the paper. What is clear from the rebuttal process is that it is hard to judge the visual fidelity alone, despite the substantial effort the authors put into the evaluation section. Especially the choice of visualizing the resulting animations via retargeting to a robot (in Blender) may have negatively impacted the reviewers' opinion. The animations in the latter part of the video are actually more convincing in my opinion. However, as pointed out before, the animations are not well-adapted to the environment, i.e., foot skating, actions on objects, etc. In future submissions on the topic, I strongly advise addressing this topic even if only partially.

**Weaknesses:**

A major question that the paper is not addressing is the generation of actions on objects and surroundings. A critical element of any character animation is to be able to specify an object acted upon, e.g., "push the red object on the table". That is not addressed at all in this paper, while it is addressed on other papers in robotics (the manipulation papers) but also animation (PADL). In PADL, a module specifically identifies the target object and animations are synthesized accordingly. In the robotics papers [Lynch2020, Stepputtis2020] the actions of the agents are also conditioned on an image of the surroundings and target objects are visually identified in the image. Potentially that could be included into InsActor through the waypoint system (maybe?), but it is so far not addressed in the paper. Personally, this is the biggest flaw of the paper, since at the moment it would not be appealing to me or many others to use this system.
Along these lines, the presented example results are not very convincing with regards to the complexity of the generated motions.

Another weakness of the paper is that it seems important details are left out or not well justified. For example, the waypoint encoding is interesting but should be explained with an example. I am particularly unsure what this would entail for the user - do we have to place the character at those waypoints and set all of its joint values? This would require the user to actually become an animator and generate keyframes. That could be a serious limitation if true. Right now I am assuming we only need to position the basic character at the waypoint (i.e. modify position and orientation of agent) not moving limbs (i.e. modify joint angles). The paper states "...replace the Gaussian noise in the first and last 25% frames with the noisy states of the character standing at the starting position and target position respectively."

Personally, I would really have liked to see a long sequence in which the agent performs a number of different actions in a row. At Siggraph they often have these cute teaser figures in which a humanoid walks, crouches, jumps etc. over a period of time.

**Questions:**

- What exactly is the transition function? I am assuming that is the differential simulator, right?

**Limitations:**

As described above, the lack interaction with the environment and linguistic goals is a major limitation that has not been brought up in the paper. The system is not able to execute commands like "lift the green cup" or "move to the lion statue". That substantially limits the applications of the approach. It is also surprising that the authors mention problems like foot skating at the beginning but do not describe how that is resolved later in the paper. While I understand that in a physics-based system the dynamics of the environment can make some of these considerations obsolete, there is still room for broken animations if this not adequately addressed. Hence, I would ask the authors to address this point in the paper.

---

> ### Author Rebuttal · Authors · 2023-08-10
>
> We thank the reviewer for comprehensive comments. We would like to address your concern as follows:
>
> > Q1: A major question that the paper is not addressing is the generation of actions on objects and surroundings. Potentially that could be included into InsActor through the waypoint system (maybe?)
>
> We agree that enabling character-object interaction is important in animation generation. Although it is a more challenging setting, we believe that our framework will be compatible with the setting by additionally including objects into the states. The high-level controller can generate motion plans conditioned on object states and the low-level controller will execute the interaction. For example, given the information of a chair, a plan for a character to sit on the chair will be generated and executed without specifically training for the task. However, extending our current waypoint encoding system to encode object information including different types, sizes, and 3D locations is highly non-trivial. In addition, the current motion data scale for human-scene interactions may not match the complexity of the problem. Therefore, we do not consider object interactions in this work. Nonetheless, we do notice recent works like [A] that show promising results on physics-based character-object interaction, and view it as a vital future work. We believe our approach will serve as an important baseline that is extendable to character-scene joint modeling and expandable with additional human-scene motion data.
>
> > Q2: Another weakness of the paper is that it seems important details are left out or not well justified. For example, the waypoint encoding is interesting but should be explained with an example.
>
> This is a great point for us to clarify! As described in the main paper, the general idea of waypoint conditioning is to encode a sequence of mean poses at the starting and the ending locations. Specifically, we first set the positions and rotations of all body links to zero, which will be a mean pose after denormalization. And then, we add a normalized position offset to move the mean pose to the starting point and the ending point. For both starting and ending, we pad the mean pose for 25% of the sequence length. In the diffusion sampling process, we fix the noise-injected mean poses following the inpainting technique. All the rest of the plan, including velocity and angular velocity in the beginning and ending state, will be generated by the diffusion planner. We will add an illustrative figure with a concrete example in future versions.
>
> > Q3: Personally, I would really have liked to see a long sequence in which the agent performs a number of different actions in a row. At Siggraph they often have these cute teaser figures in which a humanoid walks, crouches, jumps etc. over a period of time.
>
> Thanks for the great suggestion! InsActor is able to perform different actions in a row with the combination of language condition and waypoint encoding. Figure 1 included in the paper is achieved by doing so. Concretely, Fig 1. includes four text prompts which instruct the character to sequentially crouch, jump, walk, walk like a zombie to a waypoint, and finally end with a kick. We have also shown the corresponding animation in the supplementary material.
>
>
> > Q4: What exactly is the transition function? I am assuming that is the differential simulator, right?
>
> Yes, the transition function is the differentiable simulator/dynamics.
>
> Additional References:
>
> [A] Hassan et al., “Synthesizing Physical Character-Scene Interactions”, SIGGRAPH, 2023

---

> > ### Comment · Reviewer_ndRp · 2023-08-20
> >
> > I thank the authors for the detailed and careful response to both my questions and the questions of the other reviewers! I am very impressed by the deep discussions that were possible in this rebuttal. Truly appreciated!
> >
> > What is clear from the rebuttal process is that it is hard to judge the visual fidelity alone, despite the substantial effort the authors put into the evaluation section. Especially the choice of visualizing the resulting animations via retargeting to a robot (in Blender) may have negatively impacted the reviewers' opinion. The animations in the latter part of the video are actually more convincing in my opinion. More generally, I think all reviewers agree that the animations are not well-adapted to the environment, i.e., foot skating, actions on objects, etc. As a result, it seems like something is missing in this framework. Even a partial solution to this problem could have substantially added to the appeal of the paper and would have made it a clear 'accept'.
> >
> > That being said, I appreciate the effort the author put into the rebuttal - it definitely helped me better understand the nuances of the approach. I also think that (as mentioned before) the quality of the stick figure animations (second part of the video) is fairly convincing and would even be appreciated by the computer animation community.  I also acknowledge that the authors basically reimplemented the two other methods they compared against, since they did not find a public implementation. This can take a lot of time and effort.
> >
> > Based on the above, I will slightly increase my score.

---

> > > ### Author Response · Authors · 2023-08-20
> > > **Thanks to Reviewer ndRp**
> > >
> > > Thank you for raising the score! We truly appreciate your suggestions for future improvements and your acknowledgment of our efforts in establishing a systematic evaluation pipeline and re-implementing baseline approaches.

---

### Author Rebuttal · Authors · 2023-08-10

We thank reviewers for the encouragement and insightful feedback. We are glad that the reviewers found

* the problem “interesting”(R-ndRp) and “important” (R-s1C4); and
* the result “good” (R-Mekg) and “promising”(R-ndRp) and InsActor has the potential to be used in many tasks (R-K1zb); and
* the evaluation informative (R-ndRP, R-Mekg, R-K1zb, R-7eMQ); and
* the hierarchical design of InsActor “clever” (R-ndRP), “intuitive”(R-K1zb), “simple and solid” (R-s1C4).

We execute additional experiments to address reviewers' concerns. We refer the reviewers to the uploaded PDF file for more details. Experiments include:

1. Comparison between our diffusion planner and MDM. We show that our planner is built on a state-of-the-art text-to-motion generator.
2. More ablations on planning and tracking. We further verifies the performance of our planner,  show that there is a motion quality gap between plan and plan tracking, and proves the effectiveness of the tracker in our DReCon baseline.
3. Quantitative performance of low-level control. We show the training curve of our skill mapping module, which converges to a sufficiently low pose error.

We would like to address some common issues as follows:
1. Clarification of high level planner:
    * We adapted the high level planner from MotionDiffuse [40], which is on par with MDM on standard text-to-motion generation benchmarks.
    * Some visual artifacts in plans are caused by not using temporal smoothing for visualization, which has minimal impact on the final results.
    * We emphasize that in the waypoint heading setting, generating an executable plan is particularly more challenging and a compact skill mapping will be necessary.
2. Generalization of the method:
    * Generalization to human-object interaction. We acknowledge the significance of character-object interaction in animation generation. Our approach serves as an important baseline, expandable with additional object/scene modeling. We view human-object interactions as vital future work.
    * Generalization to other morphologies. Effective high-level planner training relies on ample text-motion pairs. However, data for non-human character morphologies is limited, thus, transferring to different morphologies is non trivial.
3. Tracking module details:
    * The tracking error of the low-level policy is low and is on par with previous motion tracking works.
    * Compared with standard trajectory optimization, the design of InsActor is advantageous considering computational efficiency, robustness during deployment, and flexibility for future developments.

While we agree with the importance of some limitations reviewers raised (e.g. lack of object interaction), we hope our rebuttal highlights why addressing these limitations constitute full, separate contributions themselves (e.g.,  requiring more data). We kindly ask the reviewers to let us know if further clarification or information is needed.

---

### Decision · Program_Chairs · 2023-09-21

**Decision:**

Accept (poster)

**Comment:**

The paper describes a framework for generating physically plausible human character animations from free-form text prompts. The two-stage framework first employs a diffusion model to produce high-level kinematic motion for the animated character and then utilizes a physics-based motion VAE to track this motion in a latent skill space. Experiments on the KIT-ML and HumanML3D benchmarks demonstrate that taking a hierarchical approach achieves performance gains over contemporary baselines (e.g., PADL).

The paper was evaluated by five reviewers who discussed the paper at-length with the authors and among themselves. Several reviewers found the paper to be well written and the proposed framework to be intuitive and clever. They also appreciate the focus on generating motions that are physically realizable as opposed to only considering the kinematics of the motion, which yields sound improvements over the previous state-of-the-art. However, there is also consensus among the reviewers who are concerned about the qualitative nature of the results, which demonstrate notable artifacts in the rendered motion. These concerns are particularly notable given that generating animations that are consistent with physics is a fundamental focus of the paper. There was an extensive discussion about this issue among the reviewers and between the reviewers and authors, who attribute the artifacts to issues with the differentiable Brax simulator and claim that it is secondary to the paper's primary contributions. Neither the reviewers nor the AC found these arguments to be compelling or consistent with the claims made in the paper (and its title). That said, there was agreement that the value of the paper is not necessarily in a usable system for language-enabled character animation, but rather in establishing a baseline that will foster future work in the different components of the system including differentiable simulation, motion diffusion, and physics-based tracking. The authors are strongly encouraged to temper claims about the physical plausibility of the rendered animations and to be clear about the limitations of the system, which go beyond attributing inconsistencies to issues with the chosen simulator.